# Current Status, Issues and Future Prospects of Personalized Medicine for Each Disease

**DOI:** 10.3390/jpm12030444

**Published:** 2022-03-11

**Authors:** Yuichi Yamamoto, Norihiro Kanayama, Yusuke Nakayama, Nobuko Matsushima

**Affiliations:** 1Clinical Evaluation Sub-Committee, Medicinal Evaluation Committee, Japan Pharmaceuticals Manufacturers Association, 2-3-11, Nihonbashi Honcho, Chuo-ku, Tokyo 103-0023, Japan; norihiro.kanayama@mb.kyorin-pharm.co.jp (N.K.); y2-nakayama@hhc.eisai.co.jp (Y.N.); nmatsus3@its.jnj.com (N.M.); 2Pfizer R&D Japan, 3-22-7, Yoyogi, Shibuya-ku, Tokyo 151-8589, Japan; 3Kyorin Pharmaceutical Co., Ltd., 1848, Nogi, Nogi-machi, Shimotsuga-gun, Tochigi 329-0114, Japan; 4Eisai Co., Ltd., 4-6-10 Koishikawa, Bunkyo-ku, Tokyo 112-8088, Japan; 5Janssen Pharmaceutical K.K., 3-5-2, Nishikanda, Chiyoda-ku, Tokyo 101-0065, Japan

**Keywords:** precision medicine, biomarker, rheumatoid arthritis, psoriasis, Alzheimer’s disease

## Abstract

In recent years, with the advancement of next-generation sequencing (NGS) technology, gene panel tests have been approved in the field of cancer diseases, and approaches to prescribe optimal molecular target drugs to patients are being developed. In the field of rare diseases, whole-genome and whole-exome analysis has been used to identify the causative genes of undiagnosed diseases and to diagnose patients’ diseases, and further progress in personalized medicine is expected. In order to promote personalized medicine in the future, we investigated the current status and progress of personalized medicine in disease areas other than cancer and rare diseases, where personalized medicine is most advanced. We selected rheumatoid arthritis and psoriasis as the inflammatory disease, in addition to Alzheimer’s disease. These diseases have high unmet needs for personalized medicine from the viewpoints of disease mechanisms, diagnostic biomarkers, therapeutic drugs with diagnostic markers and treatment satisfaction. In rheumatoid arthritis and psoriasis, there are many therapeutic options; however, diagnostic methods have not been developed to select the best treatment for each patient. In addition, there are few effective therapeutic agents in Alzheimer’s disease, although clinical trials of many candidate drugs have been conducted. In rheumatoid arthritis and psoriasis, further elucidation of the disease mechanism is desired to enable the selection of appropriate therapeutic agents according to the patient profile. In the case of Alzheimer’s disease, progress in preventive medicine is desired through the establishment of an early diagnosis method as well as the research and development of innovative therapeutic agents. To this end, we hope for further research and development of diagnostic markers and new drugs through progress in comprehensive data analysis such as comprehensive genomic and transcriptomic information. Furthermore, new types of markers such as miRNAs and the gut microbiome are desired to be utilized in clinical diagnostics.

## 1. Introduction

Personalized medicine refers to medical treatment that takes into account the factors such as genetic background and disease status of each individual patient and selects the most appropriate treatment and therapeutic agent. Using a patient’s genomic and transcriptomic information to diagnose diseases and predict the effects and side effects of drugs to provide highly effective treatments and therapies will improve the effectiveness of treatment. Further, it will contribute to reducing medical costs and increase the probability of success in the development of innovative new drugs, making it one of the fields with high expectations in the future.

In this article, we aimed to survey the spread of personalized medicine in disease areas other than cancer and rare diseases where personalized medicine is most advanced, in order to further promote personalized medicine in the future. The following three diseases were selected for the survey: rheumatoid arthritis, which is an inflammatory disease for which antibody drugs and JAK inhibitors are being developed; psoriasis, for which IL-17 antibodies and PDE4 inhibitors, in addition to TNFα and IL-12/23 antibodies, have been discovered in recent years, expanding the scope of treatment; and Alzheimer’s disease, for which the elucidation of disease mechanisms and the development of diagnostic methods and therapeutic agents are currently being actively pursued.

To date, the expression of specific oncogene products based on translocations of specific genes and the presence of specific mutations have been used as so-called molecular targets, such as imatinib for BCR-ABL and c-Kit mutations [1], gefitinib and erlotinib for EGFR mutations [2,3], Trastuzumab for HER2 overexpression [4], crizotinib and alectinib for ALK fusion genes [5,6], vemurafenib for BRAF mutations [7], and other driver gene aberrations and their selective inhibitors have shown high antitumor efficacy. For PD-1/PD-L1, nivolumab, and pembrolizumab, target expression and high microsatellite instability (MSI-H) are indicators of therapeutic efficacy [8]. Thus, in the field of oncology, as the number of therapeutic targets and their therapeutic agents has increased, the number of companion diagnostics has also increased. Furthermore, comprehensive analysis of the profiles of major somatic gene mutations in individual cancer patients may open the way to further personalized medicine, such as prescribing the most appropriate molecularly targeted drugs for each patient or participating in clinical trials of drugs currently under development. Oncopanel genetic testing uses NGS technology to enable comprehensive and rapid diagnosis using tumor tissue and blood samples, and attempts to select the optimal treatment for each patient based on the test results have begun. It is expected that therapeutic agents corresponding to the discovered genetic mutations will be advanced and the disease mechanism will be further elucidated [9,10]. Furthermore, such a comprehensive cancer panel test can be used to attempt to evaluate the therapeutic effect of drugs such as crizotinib and alectinib for lung adenocarcinoma on specific fusion gene-positive patients in a different carcinoma (basket study), or to select an appropriate drug from multiple investigational drugs for a specific carcinoma using the results of the panel test (umbrella trials) [11].

In the area of rare diseases, although there are various diseases, many diseases are caused by genetic background such as genetic mutation of a single gene [12]. There have been many gene therapy approaches in recent years to modify viral vectors such as adeno-associated virus (AAV) to supplement normal genes [13], and Onasemnogene abeparvovec (Zolgensma) was approved in 2019 for the treatment of spinal muscular atrophy (SMA) [14]. The nucleic acid drug Nusinersen (Spinraza) is already available in a clinical setting [15]. In order to use these drugs, genetic testing for SMN1 gene deficiency is required as a disease diagnosis, and for Zolgensma, it is also necessary to determine whether the patient has neutralizing antibodies against AAV [16]. In recent years, NGS has made it possible to analyze individual genomes more comprehensively by whole-genome sequencing (WGS) or whole-exome sequencing (WES). For undiagnosed diseases for which the causative gene has not yet been determined, NGS is desired to make a significant contribution to the progress of this research for diagnosis.

## 2. Rheumatoid Arthritis

Rheumatoid arthritis (RA) is characterized by destructive arthritis that develops against a background of an autoimmune mechanism that is very complex and heterogeneous, and without appropriate treatment, about half of patients will be bedridden after 10 years, with an average life expectancy of 10 years less. The most important factor that determines poor prognosis is joint destruction, which depends on the accumulation of disease activity. The first strategy is to control disease activity and thereby prevent joint destruction [17].

### 2.1. Diagnosis and Treatment

For the diagnosis of RA, the European League of Rheumatology (EULAR) updated their RA management recommendations in 2019 [18]. Under the criteria, when inflammation with swelling (synovitis) is observed in at least one or more joints and no disease other than RA is found to be the cause of the inflammation, the following four items are evaluated: (1) number of symptomatic joints; (2) rheumatoid factor (RF) or anti-cyclic citrullinated peptide antibody (ACPA); (3) C-reactive protein (CRP) or erythrocyte sedimentation rate (ESR); and (4) duration of symptoms. If the total score for each of these four items is six or more, RA is diagnosed and treatment with anti-rheumatic drugs is started. However, since diseases other than RA may cause the total score to be six or more, it is necessary to thoroughly examine whether other diseases are present before assigning a score.

Accurate disease activity assessment that strictly controls disease activity is essential for treatment. Currently, the most reliable evaluation method is the use of a comprehensive disease activity index [19] such as the Disease Activity Score (DAS), Simplified Disease Activity Index (SDAI), and Clinical Disease Activity Index (CDAI), which is a combination of joint findings, the Visual Analogue Scale (VAS), and CRP and ESR as inflammatory markers.

The first goal of treatment in RA is clinical remission, which is recognized as “treatment to target (T2T)” to prevent the progression of joint destruction and to maximize physical function over the long term. Methotrexate (MTX) is the mainstay drug used as first-line therapy in the absence of contraindications [20]). In patients with poor prognosis and high disease activity, a biologic TNF inhibitor may be considered in combination with MTX [21]. Other biologic agents are also emerging, such as anti-CD20 mAbs, IL-1 inhibitors, JAK inhibitors, and anti-IL-6 mAbs [22], but not all patients respond equally to these diversified agents [23], and they have many side effects. In addition, they are expensive, which imposes a heavy financial burden [24]. Therefore, personalized medicine is indispensable for prognosis, with such concerns as how to appropriately select therapeutic agents and whether it is possible to discontinue them.

### 2.2. Use of Clinical Biomarkers

Biomarkers used in the field of immunology and inflammation include acute phase reactants such as ESR and CRP, autoantibodies such as rheumatoid factor (RF), and anti-citrullinated protein antibodies (anti-CCP antibodies, ACPA) [25]. RF and ACPA are also poor prognostic factors and are useful in daily practice. In fact, there is a clear difference in disease activity and joint destruction between ACPA-positive and ACPA-negative RA, and there is a growing belief in the treatment field that biologics should be used more aggressively in the former [26].

### 2.3. Challenges for Personalized Medicine

Although the efficacy of biologics in RA is higher than that of conventional drugs [27,28], the response to treatment differs from patient to patient, as not all patients achieved remission during induction therapy. Even if remission induction is achieved it is difficult for all patients to maintain remission. In addition, only a limited number of patients can maintain remission even after maintaining remission followed by drug withdrawal [29,30].

There are many factors involved in immunity and inflammation, such as numerous immunocompetent cells and molecules including various cytokines, and no clear direction for personalized medicine has been determined at this point. In the case of biologics, there have been several reports investigating the genetic association of therapeutic drug effects, but no genetic mutations or polymorphisms have been found to provide a clear indicator. For TNFα inhibitors, studies examining genetic biomarkers have not found any indicator markers [31,32].

Some degree of genetic background has been identified for RA, and it is considered to be an other-factorial disease in which the interaction of multiple genetic and environmental factors leads to the development of the disease. Of the disease susceptibility genes identified to date, HLA is the most reliably associated across populations, and although the association between the shared epitope of HLA-DRB1 and the disease susceptibility, severity, and clinical form of RA has been established, the molecular mechanisms remain to be elucidated [33]. Furthermore, there are many reports that HLA-DRB1 correlates with ACPA-positive patients and it is desired to be used as a clinical biomarker [34].

According to reports of a GWAS meta-analysis on rheumatic diseases, most of the genetic polymorphisms associated with RA are mutations occurring in molecules related to the targets of existing rheumatic drugs and their pathways, and meta-analysis is expected to be a useful form of analysis in terms of developing new drugs and identifying new biomarkers for the treatment of rheumatic diseases [35,36].

## 3. Psoriasis

Psoriasis is a chronic inflammatory skin disease with genetic predisposition and autoimmune pathogenic features. The global prevalence is approximately 2%, with a higher prevalence in Caucasian and Scandinavian individuals as compared with Asians and Africans [37]. The main symptom of psoriasis is skin scaling (plaque) formation, but in addition to cutaneous manifestations, arthritis and fever may also develop. As for the psoriasis, there are subtypes such as psoriasis vulgaris (PsV), guttate psoriasis, erythrodermic psoriasis, generalized pustular psoriasis (GPP), and psoriatic arthritis (PsA), based on the differences in local forms and symptoms [37,38]. PsV is a representative psoriasis subtype, accounting for 85–90% of patients with psoriasis. Guttate psoriasis, erythrodermic psoriasis, and GPP are low ratio, but GPP has particularly high disease severity, and has the potential to threaten life with a pustule and high heat [39]. PsA develops with arthritis similar to rheumatism in addition to cutaneous manifestations, and accounts for 20–30% of psoriasis patients [40,41]. Therefore, it is important to ensure early diagnosis and treatment for PsA, because the irreversible joint destruction leads to a drop in quality of life (QOL) [42]. In addition to these symptoms, there are some complications in psoriasis; for example, abnormalities in nails, ocular disorders, metabolic syndrome, cardiovascular disease, type 2 diabetes, obesity, Crohn’s disease, etc., [43,44]. It also causes severe psychosocial burdens such as anxiety and depression and is a disease with low QOL [38,45].

### 3.1. Factors and Causes of Disease Onset

Psoriasis is thought to develop due to immune abnormalities associated with genetic and environmental factors [38,46]. For example, GWAS and other studies have found genes such as PSORS1-15 and other SNPs that are associated with psoriasis [37,47]. Although the pathogenesis of psoriasis is not completely understood, it is thought that some stimuli cause activation of dendritic cells and T cells and excessive production of inflammatory cytokines, leading to inflammation and abnormal proliferation of epidermal keratinocytes [48]. Cytokines involved in the pathogenesis of psoriasis include IL-17, IL-22, IL-23, TNFα, etc.

### 3.2. Diagnosis of Psoriasis

In the diagnosis of psoriasis, the Psoriasis Area and Severity Index (PASI), which is scored based on the condition of the skin (intensity of erythema, infiltration, desquamation, and range of foci), is considered to be the gold standard for assessing the severity of the disease [43,45]. Additional assessment tools are available for measuring QOL, including the Dermatology Life Quality Index (DLQI), Psoriasis Life Stress Inventory (PLSI), Psoriasis Disability Index (PDI), and Psoriasis Index of Quality of Life (PSORIQoL) [49,50]. Several genes and proteins associated with psoriasis have been found as biomarker candidates; however, there are limited validation data available to support their applications in the clinic [45,51].

### 3.3. Treatment of Psoriasis

The main treatments for psoriasis are topical therapy using ointments and phototherapy using ultraviolet irradiation, as well as systemic therapy using small molecules and biologics [37,43]. For pustular psoriasis, granulocyte and monocyte adsorption depletion therapy may also be used [52]. Treatment of psoriasis is stepwise, with topical therapy using corticosteroids, vitamin D analogues etc. for mild to moderate symptoms, and treatment based on photo therapy, oral therapy with small molecules, and biologics for moderate to severe symptoms [43]. For oral therapy, immunosuppressants such as cyclosporine and methotrexate and PDE4 inhibitors such as apremilast are used, and for treatment by biologics, antibodies such as TNFα, IL-12/23, IL-17, and IL-23 inhibitors are used [43]. These biologics are used in accordance with the patient’s symptoms [53]. For example, there are some reports that the efficacy against psoriatic arthritis tends to be higher with TNFα and IL-17 antibodies than with IL-12/23 antibodies, and IL-17 antibodies tend to aggravate the efficacy for inflammatory diseases (IBDs) [53]. The biologics were reported to have a high therapeutic effect, while 20–30% of patients have no effect [54].

### 3.4. Biomarker Research in Psoriasis

Although there are multiple subtypes of psoriasis, the pathogenesis, diagnosis, identification, and treatment method of each have not been fully established. Currently, the exploratory research for biomarkers (BMs) of psoriasis is under intense activity, and several genes, cytokines, and proteins that may be associated with the development of the disease have been identified in PsV and other subtypes [45,51,55]. For example, as related genes for psoriasis, 15 key psoriasis susceptibility loci (PSORS1-15) have been found [47]. Among them, PSORS1 is located on chromosome 6p21 within the major histocompatibility complex (MHC), which has been identified as a major susceptibility factor for psoriasis and has been attributed to 30–50% of the heritability of this disease [47]. It was also reported that PSORS1 is associated with different psoriasis subtypes, with strong associations for early-onset and guttate psoriasis, but not for late-onset (in cases occurring in individuals older than 50 years) or palmoplantar psoriasis [56]. For GPPs, recently, genetic mutations in genes encoding IL-36RN (IL-36Ra: IL-36 receptor antagonists) and CARD14, AP1S3 (Adaptor-Related Protein Complex 1 Subunit Sigma 3) were reported in relation to pustular psoriasis development [57]. GPP is also difficult to quickly distinguish from other diseases (e.g., pustules associated with infection) because there are many other diseases, such as infectious and inflammatory diseases, with similar symptoms [57]. Therefore, it is desired to develop diagnostic markers that can rapidly differentiate these diseases. Two IL-36R antibodies have shown efficacy in clinical trials, and show promise as potential therapeutic agents for new targets [58]. In psoriatic arthritis (PsA), the Disease Activity Score (DAS28) used in RA [59], the Disease Activity index for Psoriatic Arthritis (DAPSA) [60], which considers the disease state of PsA, and the ClASsification criteria for Psoriatic ARthritis (CASPAR) [43] are used as indicators of articular symptoms. However, due to the diversity of symptoms, diagnosis is difficult, and clear diagnostic criteria have not yet been established [43]. In particular, it is difficult to distinguish PsA from RA or osteoarthritis, because, in some cases, skin symptoms and arthritis develop simultaneously, and in other cases, arthritis precedes skin symptoms [40]. Since PsA is a disease that can cause serious damage to joints if diagnosis is delayed, a BM that can rapidly diagnose and predict the onset of PsA is required [42,61]. A previous report on PsA showed that the expression levels of high-sensitivity CRP, osteoprotegerin, MMP-3, and the ratio CPII of C2C differ between patients with psoriasis alone and those with PsA [62]. It was also reported that the use of biologics in PsA patients divided into four groups based on the difference in the strength of Th1 and Th17 activity in the peripheral blood resulted in a certain improvement in the therapeutic effect [63]. The discovery of new diagnostic markers and the improvement of PsA treatment in the future are desired.

## 4. Alzheimer’s Disease

Alzheimer’s disease (AD) is a progressive neurodegenerative disease characterized by cognitive decline with memory impairment and is the most common form of dementia in the elderly. The number of patients with dementia is estimated to be approximately 50 million worldwide, and as the world population ages, the number of patients with dementia continues to increase, reaching 82 million in 2030. It is projected to reach 152 million by 2050 [64].

### 4.1. Elucidation of Pathological Mechanism

In recent years, remarkable progress has been made in understanding the pathophysiology of dementia, and the essence of the disease has been discussed. The most significant trend in AD is the amyloid hypothesis [65,66]. The main component of senile plaque, which is one of the pathological features of AD, is a peptide of about 40 amino acids called amyloid β protein (Aβ). Aβ deposition is the earliest pathologically perceived lesion, Aβ aggregates and can directly exhibit neuronal toxicity, and genetic analysis of familial AD patients indicates that abnormalities in Aβ production and accumulation are deeply related to the onset of AD amyloid hypothesis.

A further neuropathological feature of AD is the intracellular accumulation of neurofibrillary tangles consisting of hyperphosphorylated tau (p-tau) as well as Aβ. This combination of amyloid (A), tau (T), and neurodegenerative (N) biomarkers is being used to classify the stage of dementia [67].

### 4.2. Gene Mutation Analysis

Most of AD is sporadic and occurs in people aged 65 years or older. It is known that sporadic AD accounts for more than half of dementia and caused by a complex interaction of environmental and genetic factors. Genetic factors play a major role in the pathogenesis of AD; it is said to be about 60 to 80%, and it is thought that many genetic factors (polygenic) are related to the pathogenesis and progression of AD. The E4 allele of the Apolipoprotein E gene (APOE) on chromosome 19 (a gene locus with a specific gene polymorphism) is known to be the highest genetic risk factor for sporadic AD [68]. One of the substances involved in the accumulation and aggregation of amyloid β peptide is apolipoprotein E. There are three main types of APOE genes, ε (epsilon) 2, ε3, and ε4, which make up a pair of two. When the relationship between the presence or absence of ε4 and the onset of AD is examined, the risk of developing AD for a genotype having one or two ε4 is about 3 to 12 times higher than that for a genotype having no ε4 [69]. However, many other genetic risk factors are still unknown.

### 4.3. Establishment of Therapeutic Agents and Treatment Methods

So far, there is no cure for AD and no way of slowing down the progression of this disease. This stems largely from our incomplete knowledge of the true pathophysiology of AD in which abnormalities of Aβ peptide and tau processing, inflammation, oxidative stress, and vascular risk factors, amongst others, are postulated to contribute [70]. Present therapeutic agents for AD consist of symptomatic therapies, which include the acetylcholinesterase inhibitors (AChEIs), such as donepezil, and the N-methyl-D-aspartate (NMDA)-receptor antagonist memantine [71]. These therapies improve the symptoms of AD, such as cognitive decline and declines in the activities of daily living and behavior, but do not alter the underlying progression of the disease. Thus, there is an urgent unmet medical need for drugs that slow or prevent the progression of the disease.

In addition, there have been many clinical trials of therapeutic drugs that radically change the progression of the disease so far, but they have not been approved as therapeutic drugs.

However, on 7 June 2021, the U.S. Food and Drug Administration (FDA) granted accelerated approval for aducanumab (aducanumab-avwa; Aduhelm™; Biogen, Solothurn, Switzerland and Eisai, Tokyo, Japan) as the first and only AD treatment to address a defining pathology of the disease, by reducing Aβ plaques in the brain [72].

On the other hand, the development of drugs targeting tau proteins is ongoing and although several clinical trials are currently underway, there are still no drugs that have been shown to be effective [73].

### 4.4. Establishment of Early Diagnosis Method

As biomarkers characteristic of AD, a decrease in the ratio of Aβ42/Aβ40 in cerebrospinal fluid (CSF) that reflects the deposition of Aβ in the brain, amyloid imaging using positron emission tomography (PET), p-tau in CSF, and tau imaging using PET, which reflect the accumulation of tau in the brain, and t-tau, NfL, etc. in CSF, which reflect neurodegeneration, are shown. However, one of the problems of PET is that the cost per examination is high and the number of facilities where the examination can be performed is limited. Furthermore, CSF is a highly invasive examination. The problem is that the throughput of the common issue between PET and CSF examinations is poor. On the other hand, in recent years, the research groups from the National Center for Geriatrics and Gerontology and Shimadzu Corporation measured Aβ-related peptides (APP669-711, Aβ1-40, and Aβ1-42) by blood tests, and compared the results with those of PET scans to determine the presence or absence of abnormal accumulation of Aβ-related peptides in the brain. It was possible to estimate with high accuracy [74]. It is desired that such low-cost, safe, and high-throughput biomarkers will be put into practical use at an early stage, and the diagnosis of dementia will be widely spread among the general public, leading to the early detection and early diagnosis of dementia [75].

## 5. Discussion

We conducted a comparison for each disease, focusing on factors that may influence personalized medicine. As the comparison object of the disease, we cited the oncology area, where personalized medicine is most advanced, and SMA, one of the rare diseases where nucleic acid drugs and gene therapy approaches have been developed in recent years (Table 1).

For the oncology area, many molecular-targeted drugs have been developed and marketed in recent years, and it is becoming possible to select more effective anticancer drugs by diagnosing the presence or absence of protein expression and gene abnormalities for each patient.

For many molecularly targeted drugs, diagnosis related to the mechanism of action of each drug allows stratification of patients who can or cannot be expected to benefit, and it is currently considered to be the disease area where personalized medicine is most prevalent. Furthermore, the high cost of treatment and the high number of side effects are also thought to contribute to the spread of personalized medicine. Similarly, in the rare diseases area, personalized medicine is advancing for diseases for which the causative genes and diagnostic methods have been clarified by the technological advances in nucleic acid drugs and gene therapy. In the case of SMA, the use of Spinraza or Zolgensma for treatment is based on the presence or absence of the target gene mutation, so testing for the presence of the gene mutation is a necessity.

Nucleic acid drugs and gene therapeutics are not readily available without diagnostic testing for the disease due to high manufacturing costs. Therefore, diagnostic tests are mandatory for the treatment of such diseases, and the drugs are used only in the patient population where they are expected to be effective. Nucleic acid drugs and gene therapy drugs can be considered as modalities for which personalized medicine is essential.

For RA area, molecular-targeted drugs have become popular in recent years, and various types of therapeutic agents, including biologics, have become available. Drugs that slow the progression of the disease, so-called DMARDs (disease-modified anti-rheumatic drugs), are the mainstream, and antibody drugs that neutralize the effects of cytokines that cause inflammation and block their inflammatory signal pathways are now available. However, for each drug, their therapeutic efficacy and side effects vary among patients, and no predictive diagnostic marker has been able to be developed (Figure 1A). The FDA’s Table of Pharmacogenomic Biomarkers in Drug Labeling also provides limited information in the RA [76]. Guidelines recommend the use of RF and ACPA as diagnostic markers of poor prognosis in rheumatoid arthritis, and they are used as indices for the selection of therapeutic agents. Although recent comprehensive genetic analyses have shown that HLA-DRB1 polymorphisms are associated with rheumatic diseases and poor prognosis, they have not yet been applied as diagnostic markers because other genetic and environmental factors may also be involved. Although treatment satisfaction is increasing, there are still some patients who do not respond well to antibody drugs and, therefore, further diagnostic biomarkers, methods of disease differentiation, and their therapeutic agents and methods are required [77]. Thus, further “personalization” and individualized therapeutic agents are required.

Similar to RA, many small molecule drugs and antibody drugs are used as therapeutic agents for psoriasis. However, compared to rheumatism, psoriasis is characterized by a greater number of subtypes and more complications. At present, the exploration of biomarkers by means of omics studies, etc. is also advanced, and it will be important for the development of personalized medicine for psoriasis to find the BM in which these subtypes and the existence of the generation of the complication are conveniently recognized. In particular, the development of markers that can accurately diagnose and predict the progression of symptoms in severe and difficult-to-treat subtypes, such as GPP and PsA, and the isolation of difficult-to-treat subtypes will lead to the development of appropriate therapies and therapeutic agents for psoriasis, as well as personalized medicine. Although patient satisfaction with psoriasis treatment has increased with the use of biologics [78], a certain percentage of patients do not respond to the medication, and further promotion of proper classification of psoriasis and appropriate drug prescription is needed (Figure 1A).

Until now, treatment for AD has focused on targeted therapy for symptoms (Figure 1B), but with the FDA approval of aducanumab (aducanumab-avwa; Aduhelm™; Biogen, Solothurn, Switzerland and Eisai, Tokyo, Japan), treatment for amyloid (A), the causative agent of dementia, will be launched. In addition, drugs for tau (T) and neurodegeneration (N), other causative agents, are desired to be developed in the future. Although testing for dementia diagnoses has generally not penetrated widely due to issues such as cost and throughput, the approval of aducanumab (aducanumab-avwa; Aduhelm™; Biogen, Solothurn, Switzerland and Eisai, Tokyo, Japan) has been driving, and the development of blood-based biomarkers will accelerate, and staging by A, T, and N that characterize individual AD pathology will likely become common with the use of a new simple and rapid method.

In the future, it is hoped there will be progress to evaluate and identify the histopathology of individual dementia in a multifaceted manner, and tailored treatment tailored to the pathology of individual dementia will be implemented.

## 6. Future Perspectives

In the oncology area, further promotion of personalized medicine using genomic panel diagnosis is expected. With these diagnoses, there are still about 10–20% of patients applicable for therapeutic drug selection and ongoing clinical trial participation, with clinical benefit for approximately half of these patients [79,80]. They are expected to quickly provide patients with drugs that are predicted to be more effective, thereby further enhancing the effectiveness of treatment. This will also reduce the cost of medical care and the burden of treatment on the patient [80]. However, there are limited data that clearly show the improvement in the quality of life for patients in actual clinical practice [81]. Further research and development of new therapeutic agents corresponding to further genetic analysis progress and diagnosis is awaited. In recent years, many early diagnosis attempts using miRNA extracted from exosomes in peripheral blood have also been reported [82], and efforts have been made toward practical application.

In RA and psoriasis, the causes of the diseases are still unknown, and the mainstream of therapeutic drugs are disease-modifying drugs that aim to improve symptoms or prevent deterioration. For both diseases, further elucidation of the complex involvement of genetic backgrounds, environmental factors, and the immune system in the pathogenesis of the diseases is awaited. Regarding the elucidation of diseases, detailed classifications, and the study and development of innovative new drugs, we would like to desire future advances in genetic analysis, comprehensive gene expression and protein expression analysis, epigenetic analysis, genomic analysis of the gut microbiome, and the research and development of novel biomarkers such as miRNA. It is hoped that this will lead to new drug research, the development of new concepts, and the development of diagnostic markers for therapeutic drugs (Figure 1C). In the future, the collection and storage of Biobank samples will also be crucial for the study and development of diagnostic markers and should be considered as a tool to explore patient-stratified markers and to elucidate disease MOAs, such as by Translational research and exhaustive post-hoc analysis. In addition, for diagnostic methods that involve new technologies, it is necessary to consider the reducing of diagnostic costs for their widespread use. Acceptable diagnostic costs depend on the cost and cost-effectiveness of the treatment of the target disease. For example, it was reported that the cost of a test for the early diagnosis of rheumatoid arthritis (RA) should not exceed €200–€300 [83].

On the other hand, there are a number of diseases, such as Alzheimer’s disease, that are difficult to treat even if they are diagnosed after the disease has progressed to a certain degree. Therefore, it is necessary to consider early diagnosis (treatment) or preventive medicine (prediction and forecasting) as one of the ways to view personalized medicine. The concept of personalized healthcare can be considered as the alternative. It is also important to promote preventive medicine without drugs and early diagnosis [84].

It is hoped that cohort studies from the early stages of the disease and advances in comprehensive data collection and data analysis techniques for clinical biomarkers will further stimulate the elucidation of the mechanisms of disease progression, the development of preventive and curative measures to control the onset of disease, and the research and development of new types of therapeutic agents.

## Figures and Tables

**Figure 1 jpm-12-00444-f001:**
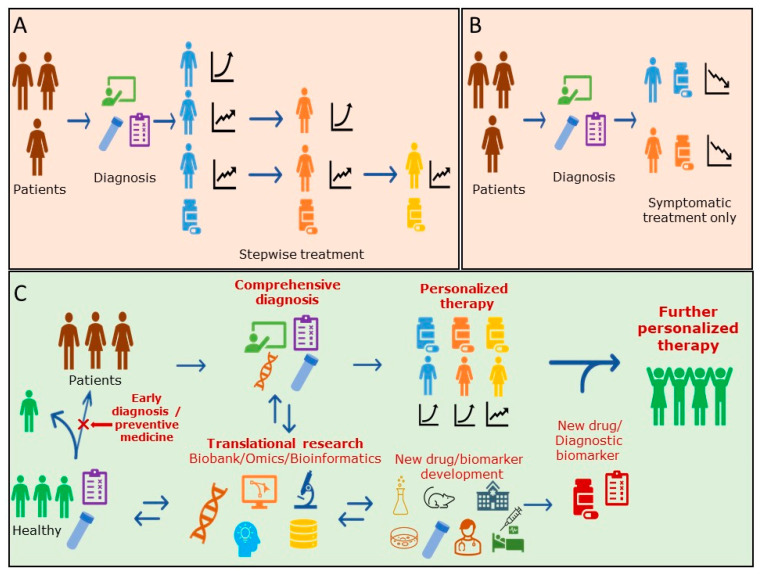
Summary diagrams for patient treatment regimen for current RA and Psoriasis (**A**), Alzheimer’s disease (**B**) and the scheme of future personalized therapy (**C**).

**Table 1 jpm-12-00444-t001:** Comparison of items related to personalized medicine for each disease.

Item	Cancer	Rheumatoid Arthritis	Psoriasis	Alzheimer’s Disease	Spinal Muscular Atrophy (SMA)
Understanding of Disease MOA	Somatic mutation of driver genes (monogenic)	autoimmune disease, polygenic	autoimmune disease, polygenic	polygenic	SMN1 gene mutation
Diagnostic markers(test)	numerous (Oncogene panel tests)	RF, ACPA	-	PET (Aβ42/Aβ40, tau, Neurodegeneration)	SMN1 gene mutation
Drug (CDx)	Herceptin (HRE2) Gefitinib (EGFR) Imatinib (BCL-Abl) Crizotinib(ALK fusion) Nivolumab (PD-1, MSI) etc.	-	-	-	Onasemnogene abeparvovec (Anti-AAV9) Nusinersen (SMN2 gene copy number)
Treatment satisfaction	Low-medium	medium	medium	low	low-medium

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
