# Peer review of "Current Status, Issues and Future Prospects of Personalized Medicine for Each Disease"

_jpm, 2022, doi:10.3390/jpm12030444_

Round 1

Reviewer 1 Report

Dear Authors,

Please provide some details about the comments below to improve the quality of the review article:

  1. How many patients actually respond to the identified targeted therapy by screening of using sequencing to make the identified patients eligible for targeted therapy?
  2. Please mentioned some details about the cost of testing and therapy?
  3. Please mention about the quality of life of the patients after taking these targeted therapy.
  4. Please write about the benefits to the patients of the targeted therapy.
  5. What are the alternatives to current PM efforts?

Author Response

Dear Reviewer

Many thanks for reviewing our manuscript and providing the comments and I apologize for the delay in the response.

They are very helpful comments to improve the manuscript.

Our responses to your comments are below and the attached file is the revised manuscript. Could you confirm if our responses meet your requests?

Thank you and regards,

Yuichi

1. How many patients actually respond to the identified targeted therapy by screening of using sequencing to make the identified patients eligible for targeted therapy?

The information below was added to Line 438-445 in the revised manuscript.

According to the references below, Results using MSK-impact indicate that approximately 10-20% of subjects are amenable to targeted therapies, with clinical benefit for approximately half of these patients. 

Reference

Jordan EJ, Kim HR, Arcila ME, Barron D, Chakravarty D, Gao J, Chang MT, Ni A, Kundra R, Jonsson P, Jayakumaran G, Gao SP, Johnsen HC, Hanrahan AJ, Zehir A, Rekhtman N, Ginsberg MS, Li BT, Yu HA, Paik PK, Drilon A, Hellmann MD, Reales DN, Benayed R, Rusch VW, Kris MG, Chaft JE, Baselga J, Taylor BS, Schultz N, Rudin CM, Hyman DM, Berger MF, Solit DB, Ladanyi M, Riely GJ.Prospective Comprehensive Molecular Characterization of Lung Adenocarcinomas for Efficient Patient Matching to Approved and Emerging Therapies.. Cancer Discov. 2017 Jun;7(6):596-609. doi: 10.1158/2159-8290.CD-16-1337. Epub 2017 Mar 23.

Soumerai TE, Donoghue MTA, Bandlamudi C, Srinivasan P, Chang MT, Zamarin D, Cadoo KA, Grisham RN, O'Cearbhaill RE, Tew WP, Konner JA, Hensley ML, Makker V, Sabbatini P, Spriggs DR, Troso-Sandoval TA, Charen AS, Friedman C, Gorsky M, Schweber SJ, Middha S, Murali R, Chiang S, Park KJ, Soslow RA, Ladanyi M, Li BT, Mueller J, Weigelt B, Zehir A, Berger MF, Abu-Rustum NR, Aghajanian C, DeLair DF, Solit DB, Taylor BS, Hyman DM. Clinical Utility of Prospective Molecular Characterization in Advanced Endometrial Cancer.. Clin Cancer Res. 2018 Dec 1;24(23):5939-5947. doi: 10.1158/1078-0432.CCR-18-0412. Epub 2018 Aug 1

2. Please mentioned some details about the cost of testing and therapy?

The sentences below were added to line 465-468. However, our manuscript didn’t focus on the details for the cost of testing and therapy. It is very important discussion, however, it was challenging issue for us to discuss.

Acceptable diagnostic costs depend on the cost and cost-effectiveness of the treatment of the target disease. For example, it has been reported that the cost of a test for early diagnosis of rheumatoid arthritis (RA) should not exceed €200- €300

3. Please mention about the quality of life of the patients after taking these targeted therapy.

The information below was added to Line 438-445 in the revised manuscript.

 In the case of targeted drugs with diagnostics, it is desired to improve the patient's quality of life, i.e., reintegration into society and return to normal daily activities. The results of clinical trials have shown the efficacy of drugs, and in the case of cancer, the efficacy has been shown in terms of PFS and OS. However, there are limited data that clearly show the improvement in the quality of life of patients in actual clinical practice.

Reference

Haslam A, Herrera-Perez D, Gill J, Prasad V. Patient Experience Captured by Quality-of-Life Measurement in Oncology Clinical Trials. JAMA Netw Open. 2020 Mar 2;3(3):e200363. doi: 10.1001/jamanetworkopen.2020.0363. 

4. Please write about the benefits to the patients of the targeted therapy.

The information below was added to Line 438-445 in the revised manuscript.

In the case of molecularly-targeted drugs accompanied by diagnostic agents, it is expected to provide patients with drugs that are predicted to be more effective more quickly and appropriately, thereby further enhancing the effectiveness of treatment. This will also reduce the cost of medical care and the burden of treatment on the patient.

Reference

Soumerai TE, Donoghue MTA, Bandlamudi C, Srinivasan P, Chang MT, Zamarin D, Cadoo KA, Grisham RN, O'Cearbhaill RE, Tew WP, Konner JA, Hensley ML, Makker V, Sabbatini P, Spriggs DR, Troso-Sandoval TA, Charen AS, Friedman C, Gorsky M, Schweber SJ, Middha S, Murali R, Chiang S, Park KJ, Soslow RA, Ladanyi M, Li BT, Mueller J, Weigelt B, Zehir A, Berger MF, Abu-Rustum NR, Aghajanian C, DeLair DF, Solit DB, Taylor BS, Hyman DM. Clinical Utility of Prospective Molecular Characterization in Advanced Endometrial Cancer.. Clin Cancer Res. 2018 Dec 1;24(23):5939-5947. doi: 10.1158/1078-0432.CCR-18-0412. Epub 2018 Aug 1

5. What are the alternatives to current PM efforts?

 Line 469-474 paragraph was modified according to the reference below.

The alternatives to current PM effort are considered to be early diagnosis/prediction before suffering from disease and preventive medicine/treatment. And personalized healthcare is also important for the patients using drugs such as alleviators even if there is no appropriate treatment.

Reference

 Anna Pokorska-Bocci, Alison Stewart, Gurdeep S Sagoo, Alison Hall, Mark Kroese, Hilary Burton. 'Personalized medicine': what's in a name? Per Med. 2014. 11(2), 197-210. DOI: 10.2217/pme.13.107

Reviewer 2 Report

I have reviewed the manuscript entitled “Current status, issues and future prospects of personalized medicine for each disease” by Yuichi Yamamoto and the coauthors. This manuscript has been submitted as review and largely provides an adequate summary of the research progress made in personalized medicine area related to three important medical conditions, Rheumatoid Arthritis and Psoriasis as the inflammatory diseases, and Alzheimer's disease. Here I have the following comments.

General comments.

  1. A table/ diagram describing a summary of the patient management regime currently underway in the clinics should be included. For instance, how the first line of therapy is determined and followed by categorizing patients between responders and non-responders. How non-responders are then prioritized for other treatment regiments and so on and so forth …. This will help the developing clinician readers to see a quick overview of how patients are currently managed in the clinic.
  2. Likewise, the authors are suggested to make a diagram about various diagnostic techniques and procedures currently adopted to establish initial diagnosis in the case of these three diseases, and how genetics, epigenetic and pharmacogenetics studies and tools are expected to pave the way for personalized medicine for better treatment. A schematic figure perhaps with three panels of flow diagrams corresponding to each disease in the future direction section would lead the readers into the future of the personalized medicine.
  3. In the current format of the manuscript, the text contains too long sentences, which, in many cases brought significant unclarity to the critical messages that the authors want to get across, and as such it is hard to follow (as examples of long sentences, abstract lines: 22-26 and lines: 33-37; Introduction lines: 43-48; and in section 2.3. challenges to personalized medicine, page: 4, lines: 158-163). The authors should shorten all the sentences exceeding three lines into two sentences through the manuscript.

Specific comments:

  1. Please try to shorten three cancer and rare disease related paragraphs in the introduction and start giving in depth background of why personalized medicine approaches are necessary in the future to cater these disease in the clinic.  
  2. Please change the “genome information and gene expression information” in introduction on page: 1, line: 43-44 into genomic and transcriptomic information.
  3. Please change the sentence in inverted commas “Several genes and proteins associated with psoriasis have been found as biomarker candidates, however there are limited validation data to support them use in clinical [48] [54]” on page: 5, lines: 222-224 with the under lined one Several genes and proteins associated with psoriasis have been found as biomarker candidates, however there are limited validation data available to support their applications in the clinic.

Author Response

Dear Reviewer

Many thanks for reviewing our manuscript and providing the comments and I apologize for the delay in the response.

They are very helpful comments to improve the manuscript.

Our responses to your comments are below and the attached file is the revised manuscript. Could you confirm if our responses meet your requests?

Thank you and regards,

Yuichi

General comments.

1. A table/ diagram describing a summary of the patient management regime currently underway in the clinics should be included. For instance, how the first line of therapy is determined and followed by categorizing patients between responders and non-responders. How non-responders are then prioritized for other treatment regiments and so on and so forth …. This will help the developing clinician readers to see a quick overview of how patients are currently managed in the clinic.

Diagram A and B in Figure 1 was putted in the discussion section in the revised manuscript

2. Likewise, the authors are suggested to make a diagram about various diagnostic techniques and procedures currently adopted to establish initial diagnosis in the case of these three diseases, and how genetics, epigenetic and pharmacogenetics studies and tools are expected to pave the way for personalized medicine for better treatment. A schematic figure perhaps with three panels of flow diagrams corresponding to each disease in the future direction section would lead the readers into the future of the personalized medicine.

Diagram C in Figure 1 was putted in the discussion section in the revised manuscript

3. In the current format of the manuscript, the text contains too long sentences, which, in many cases brought significant unclarity to the critical messages that the authors want to get across, and as such it is hard to follow (as examples of long sentences, abstract lines: 22-26 and lines: 33-37; Introduction lines: 43-48; and in section 2.3. challenges to personalized medicine, page: 4, lines: 158-163). The authors should shorten all the sentences exceeding three lines into two sentences through the manuscript.

The long sentences in the manuscript were shorten accordingly.

Specific comments:

1. Please try to shorten three cancer and rare disease related paragraphs in the introduction and start giving in depth background of why personalized medicine approaches are necessary in the future to cater these disease in the clinic. 

The cancer and rare disease related paragraph were shorten and the background of the purpose of this review article was moved to earlier in the introduction section.

We basically think that personalized medicine approaches are necessary for all disease area, however, different approaches should be considered for each disease area. So we investigated the current status of the diagnosis and treatment in the three disease areas and their researches for personalized medicine.

2. Please change the “genome information and gene expression information” in introduction on page: 1, line: 43-44 into genomic and transcriptomic information.

The sentence was changed accordingly.

3. Please change the sentence in inverted commas “Several genes and proteins associated with psoriasis have been found as biomarker candidates, however there are limited validation data to support them use in clinical [48] [54]” on page: 5, lines: 222-224 with the under lined one Several genes and proteins associated with psoriasis have been found as biomarker candidates, however there are limited validation data available to support their applications in the clinic.

The sentence was changed accordingly.

Round 2

Reviewer 1 Report

Dear Author,

Thank you for revising your manuscript on the basis of comments provided.